# Effects of Hydrophobic Gold Nanoparticles on Structure and Fluidity of SOPC Lipid Membranes

**DOI:** 10.3390/ijms241210226

**Published:** 2023-06-16

**Authors:** Poornima Budime Santhosh, Tihomir Tenev, Luka Šturm, Nataša Poklar Ulrih, Julia Genova

**Affiliations:** 1Institute of Solid State Physics, Bulgarian Academy of Sciences, Tzarigradsko Chaussee 72, 1784 Sofia, Bulgaria; tenev@issp.bas.bg (T.T.); julia.genova@issp.bas.bg (J.G.); 2Department of Food Science and Technology, Biotechnical Faculty, University of Ljubljana, 1000 Ljubljana, Slovenia; luka.sturm@bf.uni-lj.si (L.Š.); natasa.poklar@bf.uni-lj.si (N.P.U.)

**Keywords:** gold nanoparticles, phospholipids, biomedical applications, membrane fluidity, infrared spectroscopy

## Abstract

Gold nanoparticles (AuNPs) are promising candidates in various biomedical applications such as sensors, imaging, and cancer therapy. Understanding the influence of AuNPs on lipid membranes is important to assure their safety in the biological environment and to improve their scope in nanomedicine. In this regard, the present study aimed to analyze the effects of different concentrations (0.5, 1, and 2 wt.%) of dodecanethiol functionalized hydrophobic AuNPs on the structure and fluidity of zwitterionic 1-stearoyl-2-oleoyl-*sn*-glycerol-3-phosphocholine (SOPC) lipid bilayer membranes using Fourier-transform infrared (FTIR) spectroscopy and fluorescent spectroscopy. The size of AuNPs was found to be 2.2 ± 1.1 nm using transmission electron microscopy. FTIR results have shown that the AuNPs induced a slight shift in methylene stretching bands, while the band positions of carbonyl and phosphate group stretching were unaffected. Temperature-dependent fluorescent anisotropy measurements showed that the incorporation of AuNPs up to 2 wt.% did not affect the lipid order in membranes. Overall, these results indicate that the hydrophobic AuNPs in the studied concentration did not cause any significant alterations in the structure and membrane fluidity, which suggests the suitability of these particles to form liposome–AuNP hybrids for diverse biomedical applications including drug delivery and therapy.

## 1. Introduction

Engineered nanomaterials (ENMs) with a particle size between 1 and 100 nm are widely used in various sectors including food, energy, and cosmetics [1,2]. Among the different types of ENMs, gold nanoparticles (AuNPs) have emerged as an excellent candidate for numerous biomedical applications including sensors, imaging, drug delivery, and therapeutics due to their facile surface chemistry, ease of synthesis, tunable size, and unique localized surface plasmon resonance properties [3]. Due to their ability to absorb near-infrared light strongly at specific wavelengths and produce heat to destroy cancer cells, AuNP-mediated photothermal therapy (PTT) is an effective approach in cancer treatment [4]. Further, the AuNPs conjugated with anticancer drugs have the potential to produce a synergetic chemo-photothermal effect to drastically improve the therapeutic efficacy [5]. However, the extensive use of AuNPs in the biomedical field has raised serious concerns regarding their cytotoxic effects and impact on human health.

In general, when the nanoparticles approach a cell, they interact first with the cell membrane, which acts as a barrier for their entry inside a cell [6]. One of the key events concerned with the cytotoxicity of nanoparticles is their interaction with various components such as lipids and proteins in the cell membrane. Therefore, investigating the effects of different types of ENMs on various biophysical properties and functions of cell membranes is crucial in advancing the industrial and biomedical applications of nanotechnology. Since the cell membrane is a complex structure, it is difficult to study the various mechanisms of nanoparticle–membrane interactions. Hence, lipid vesicles or liposomes, which are closed, quasi-spherical, self-assembled lipid bilayer structures are widely used as the simplest model system to study the influence of nanoparticles. Liposomes can be formed from both synthetic and natural lipids using different experimental techniques in laboratory conditions [7]. Furthermore, the composition of these lipid membranes can be manipulated by incorporating different proteins, carbohydrates, and cholesterol molecules to mimic the real cell membranes [8]. It is worth noticing that liposomes are versatile drug delivery systems and are used as carriers in several approved medicaments in cancer therapeutics [9,10].

It is well known that the rapid interactions of nanoparticles with lipid membranes can cause membrane damage or alter the membrane properties such as stability, fluidity, elasticity, and permeability [11,12,13]. The interactions of AuNPs with different types of phospholipid membranes have been previously reported using molecular dynamics simulations and by investigating specific membrane properties [14,15,16,17,18]. However, a detailed study using the combination of fluidity measurements and structural analysis of hydrophobic AuNPs on the zwitterionic 1-stearoyl-2-oleoyl-*sn*-glycerol-3-phosphocholine (SOPC) phospholipid membranes has not been previously reported, to the best of our knowledge. Taking this into consideration, the present study aims to investigate the effects of different concentrations (0.5, 1, and 2 wt.%) of dodecanethiol-functionalized hydrophobic AuNPs on the structure and fluidity of SOPC lipid bilayer membranes using Fourier-transform infrared (FTIR) spectroscopy, and fluorescent spectroscopy, respectively. The findings of this study can help to understand the effects of AuNP–membrane interactions, which is crucial to gain fundamental knowledge of the main determinants driving the phenomena at the nano–bio interface and to design novel functional liposome–AuNP hybrids for biomedical applications.

## 2. Results and Discussion

### 2.1. Nanoparticles Characterization

The core size of AuNPs was found to be 2.2 ± 1.1 nm using transmission electron microscopy (TEM) (Figure 1A). Dynamic light scattering (DLS), which is a sensitive technique to assess the aggregation and stability of nanoparticles, was used to measure the size distribution of AuNPs (Figure 1B).

The DLS data showed a slight increase in the size of AuNPs (3.5 ± 1.6 nm) when compared to the TEM size. This is because TEM measured the actual particle size, whereas DLS indicated the hydrodynamic diameter and measured the thickness of the surface coating and solvent layer surrounding the nanoparticles [19].

### 2.2. Membrane Fluidity

Membrane fluidity indicates the viscosity of the lipid bilayer. The changes in viscosity can affect the rotation and diffusion of proteins and other biomolecules within the membrane. Therefore, maintenance of optimal membrane fluidity is essential for the cell to perform diverse functions such as diffusion of small molecules, cell signaling, and fusion [11,14]. Moreover, several pathological processes can also be related to fluidity modifications [20]. The membrane fluidity of multilamellar vesicles (MLVs) prepared with SOPC lipid (Figure 2A) was analyzed through temperature-dependent anisotropy measurements by employing two fluorescent probes, 1, 6-diphenyl-1,3,5-hexatriene (DPH) (Figure 2B) and its cationic derivative 1-(4-trimethylammoniumphenyl)-6-phenyl-1,3,5-hexatriene (TMA-DPH) (Figure 2C).

DPH and TMA-DPH are widely used to study the membrane dynamics and architecture in real cells, as well as in artificial lipid vesicles [20,21,22]. DPH, being a hydrophobic probe, localizes in the hydrophobic tail region of the lipids, whereas the hydrophilic probe TMA-DPH anchors near the head region of lipids at the water–lipid interface [21]. Therefore, simultaneous use of these probes is an ideal method to measure the viscosity changes at both the head and the tail regions of the lipids. The fluorescent spectroscopic technique enables to measure the anisotropy values of DPH and TMA-DPH, which depends on the packing order of lipid chains in the membrane. Hence, the anisotropy measurements are directly proportional to the order parameter values and inversely proportional to the membrane fluidity [22]. Since the present study intends to investigate and correlate the behavior of hydrophobic AuNPs with real cell membranes for potential biomedical applications, the temperature interval for fluidity measurements was chosen between 15 °C and 55 °C, as temperatures below and above this range do not have much clinical significance. In general, increasing the temperature increases the membrane fluidity, as the fatty acid tails become less rigid, and the phospholipids gain enough kinetic energy to overcome the intermolecular forces that hold the membrane lipids together [23]. The lipid order parameter values were determined for SOPC MLVs with different concentrations of AuNPs using two fluorescent probes, DPH (Figure 3A) and TMA-DPH (Figure 3B). Membrane fluidity was evaluated by comparing the order parameter values in pure SOPC MLVs (control) and those with entrapped AuNPs.

As shown in Figure 3A, the initial order parameter value for pure SOPC MLVs using DPH at 15 °C was 0.52 ± 0.01, while those for SOPC MLVs with 0.5, 1, and 2 wt.% AuNPs were found to be 0.519 ± 0.01, 0.510 ± 0.01, and 0.51 ± 0.01, respectively. As the temperature was gradually increased, the order parameter values slightly decreased and the final order parameter values for the control and SOPC MLVs entrapped with 0.5, 1, and 2 wt.% AuNPs at 55 °C were found to be 0.167 ± 0.01, 0.162 ± 0.01, 0.164 ± 0.01, and 0.167 ± 0.01, respectively. The initial and final order parameter values measured at 15 and 55 °C for all the samples using DPH and TMA-DPH probes are shown in Table 1.

Schachter et al. [24] reported that the anisotropy values are usually high in the gel phase of the lipids, decrease in the liquid-disordered state as the temperature increases, and reach intermediate values in the liquid-ordered state. Accordingly, our results showed a steady decrease in order parameter values of all the samples and a gradual increase in the membrane fluidity, as the temperature was increased. However, it is critical to understand that, when compared to the control values, the order parameter values of SOPC MLVs entrapped with different concentrations of AuNPs were almost identical. These results infer that the AuNPs did not show a major impact on the membrane fluidity, probably due to the extremely small size (2.2 nm) and low concentrations of AuNPs used in this study. When the concentration of AuNPs was increased to 5 wt.% in the MLVs, it impaired the vesicle formation as the incorporation of more AuNPs inside the lipid bilayer disrupted the membrane. Consequently, a lower concentration of AuNPs up to 2 wt.% was chosen in this study. These results are significant as they reveal that care has to be taken when choosing the appropriate concentration of nanoparticles, especially while designing nanoparticle-based drug delivery systems; otherwise, higher concentrations of nanoparticles may cause membrane rupture or damage and lead to adverse effects.

Similar results were obtained in our previous work, where we reported that plain superparamagnetic iron oxide nanoparticles (SPIONs) with a TEM size of 11 nm and surface-functionalized SPIONs (size: 20 nm), either incubated or encapsulated inside the liposomes, did not show any considerable effect on membrane fluidity and phase transition of 1,2-dipalmitoyl-*sn*-glycero-3-phosphocholine (DPPC) liposomes [25]. Our results also coincide with the findings of Park et al. [26], who investigated the fluidity of DPPC lipid bilayers loaded with AuNPs (size: 3–4 nm) using DPH fluorescence anisotropy measurements. Their results showed that the AuNPs induced slight fluidity modulations, which were attributed to an increase in the temperature and the interaction of AuNPs with lipid molecules in the bilayer. Mhashal et al. [14] used all-atomistic molecular dynamics simulations to study the effect of single AuNP interaction on the fluidity of membranes prepared with 1-arachidoyl-2-oleoyl-*sn*-glycero3-phosphocholine (AOPC) lipid. The simulation results showed that the lipid molecules located near the site of AuNP interacted directly with them, leading to membrane deformation. However, lipid molecules located far away from the interaction site of the AuNPs became perturbed, which induced alterations in the local ordering of the lipid domains and in bilayer thickness. When the size of AuNPs (2 to 5 nm) was changed, a similar trend, but with a different magnitude of lipid order, was observed.

The size, shape, and surface chemistry of the nanoparticles are important parameters that govern their interactions with lipid membranes, cellular uptake, and the associated toxic effects [25,26,27,28]. To understand this effect, Contini et al. [15] investigated the size-dependent interaction of citrate-stabilized AuNPs (5 to 60 nm) on membranes prepared with 1-palmitoyl-2-oleoyl-*sn*-glycero-3-phosphocholine (POPC) and 1,2-dileoyl-*sn*-glycero-3-phosphocholine (DOPC) lipids. On the basis of the ratio of vesicle and nanoparticle area, the AuNPs either self-assembled or interacted with the membrane lipids in a different fashion. TEM images showed that the smaller AuNPs (5–10 nm) either formed aggregates on the outer surface of the membrane or were engulfed by the formation of wrapped linear aggregates within a tubular membrane. Conversely, medium-size AuNPs (25–35 nm) were adsorbed on the bilayer surface and induced membrane bending with an observable penetration depth. The adsorption process of larger AuNPs (50–60 nm) was disturbed by enhanced membrane tension owing to the reduced liposome/AuNPs surface area ratio. Taken together, these studies report that AuNPs with different morphologies and surface chemistry interact with lipid membranes in different fashions and cause variations in the structure and fluidity.

In the case of TMA-DPH (Figure 3B), the order parameter values for pure SOPC MLVs (control) and the MLVs entrapped with 0.5, 1, and 2 wt.% AuNPs at 15 °C were found to be 0.79 ± 0.01, 0.79 ± 0.01, 0.80 ± 0.01, and 0.80 ± 0.01, respectively. At 55 °C the order parameter values for the same samples decreased to 0.71 ± 0.01, 0.68 ± 0.01, 0.70 ± 0.02, and 0.69 ± 0.01, respectively. Similar to DPH results, the order parameter for the SOPC MLVs entrapped with different concentrations of AuNPs was almost identical to the control values. Since TMA-DPH indicates the lipid order at the water–lipid interface, no drastic changes are usually expected near the head region of lipids [29,30]. Accordingly, our results also showed no major differences in lipid order parameter values among all samples at the water–lipid interface. Overall, both DPH and TMA-DPH anisotropy measurements showed that, with the gradual increase in the temperature, the order parameter values decreased steadily in all the samples with a proportional increase in the membrane fluidity. However, it is worth noting that the difference in the lipid order parameter values between control vesicles and MLVs entrapped with AuNPs was almost negligible, which indicates that the presence of hydrophobic AuNPs in the SOPC MLVs did not show any significant impact on the membrane fluidity, at least not within the studied temperature range and percentage of incorporated AuNPs. These results also coincide with our previous reports, where we studied the influence of calcium ions, iron oxide, and cobalt ferrite nanoparticles on the fluidity and bending elasticity of lipid membranes prepared with different phospholipids and archaeal lipids [31,32,33,34,35,36].

### 2.3. FTIR Spectroscopy

FTIR is a widely used technique in biomedical research, as it enables the precise analysis of various biomolecules such as lipids, proteins, and nucleic acids, without the need for external labels or tedious sample preparation methods [37,38,39,40]. It works on the principle that different molecules absorb light and vibrate at specific frequencies, which is characteristic of the chemical bonds present. FTIR is a sensitive technique, and the presence of admixtures such as nanoparticles and their strong interactions with lipid molecules will induce notable changes in different vibrational modes of the IR spectrum [41]. Therefore, the IR spectrum helps to identify the different functional groups in the sample and provides a molecular fingerprint for the structural analysis. Literature reports have shown that several spectroscopic features can be used to monitor lipid phase transitions, such as (i) the shift of C–H symmetric and antisymmetric stretching frequency, (ii) the shift of phosphate antisymmetric stretching frequency, and (iii) the stretching, scissoring, rocking, wagging, and twisting vibrational modes of methylene moieties of lipid chains [42,43,44].

In this study, FTIR data were used to analyze the effect of different concentrations of AuNPs on the dynamics and structure of SOPC lipids by comparing the wavenumber shifts of various vibrational modes corresponding to the lipid heads, acyl chains, and the interfacial region. The antisymmetric and symmetric stretching vibrations of both the methyl (CH_3_) and methylene (CH_2_) groups of phospholipids are predominantly observed in the 3050–2800 cm^−1^ spectral range, which provides information about changes in the conformation of lipid chains. The functional groups generally focused in the 1800–1000 cm^−1^ spectral range include the carbonyl (C=O) stretching mode (1738 cm^−1^) and phosphate (PO_2_^−^) antisymmetric stretching mode (1222 cm^−1^), which indicates the structural changes at the interfacial and head region of lipids, respectively [45]. Hence, these spectral regions were carefully investigated in the present study to detect the characteristic vibrations from different regions of the lipid molecules.

FTIR spectra were recorded for pure SOPC MLVs and those entrapped with 0.5, 1, and 2 wt.% AuNPs (Figure 4). The FTIR spectra obtained for SOPC MLVs showed a specific vibration pattern that is typical for chemical groups present in phospholipids. The (C–H_3_)_3_N^+^ antisymmetric vibration observed at 3037 cm^−1^ and the low-frequency vibration of the methyl groups at 969.5 cm^−1^ were attributed to the antisymmetric C–N–C stretch in the choline group of lipids. A similar frequency vibration corresponding to the choline group of SOPC lipids was observed in our previous studies, where we used FTIR data for the structural analysis of SOPC MLVs incorporated with different proportions of cholesterol and melatonin hormone [46,47].

#### 2.3.1. Analysis of Symmetric and Antisymmetric C–H Vibrations

FTIR spectra of SOPC MLVs entrapped with AuNPs revealed slight changes in the peak intensities and band shift when compared to the pure SOPC MLVs. For instance, the frequency values of CH_2_ antisymmetric stretching bands of pure SOPC MLVs decreased slightly from 2923 to 2922.5, 2922, and 2921 cm^−1^ with a gradual increase in the AuNP concentration from 0.5, 1, and 2 wt.%, respectively. The frequency values of CH_2_ symmetric stretching bands also showed a slight decrease from 2850 cm^−1^ (control value) to 2848.5 cm^−1^ as the AuNP concentration increased to 2 wt.%, suggesting that the AuNPs slightly disturbed the acyl chain flexibility. However, no drastic peak shifts or band intensities were observed, which is consistent with our fluidity data. These results coincide with the findings of Krecisz et al. [48] who reported the effect of polymer-coated iron oxide-based magnetic nanoparticles (MNPs) with different core sizes (3, 10, and 13 nm) on liposomes prepared using 1,2-dimyristoyl-*sn*-glycero-3-phosphocholine (DMPC) phospholipid. In liposomes incorporated with 10 nm and 13 nm MNPs, the wavenumbers characterizing the CH_2_ antisymmetric stretching band decreased slightly from 2923 to 2921 cm^−1^, but they did not show any impact on the main phase transition temperature of DMPC lipids, indicating no interaction of the MNPs with hydrophobic tails of the lipids. Since the symmetric and antisymmetric vibrations at 2800–3000 cm^−1^ derived from the CH_2_ and CH_3_ groups of the acyl chains indicate the conformational changes within the lipid bilayer, analyzing this spectral region can provide insight into AuNP–membrane interactions in the hydrophobic tail region of lipids. The various frequency shifts induced by AuNPs in comparison with the pure SOPC MLVs in the measured spectral region are summarized in Table 2.

#### 2.3.2. Analysis of Carbonyl and Phosphate Group Vibrations

Since the frequency of the carbonyl group absorption band strongly relies on the hydration state of the lipid headgroups, it is a sensitive reporter to probe structural variations in the headgroup environment [49]. Hence, the C=O stretching vibrations from the ester group of phospholipids were analyzed to derive information about the interaction of AuNPs with the polar headgroup regions of lipids at the interfacial region. The FTIR spectra showed that the C=O stretching frequency at 1738.2 cm^−1^ for pure SOPC MLVs was slightly reduced to 1737.3 cm^−1^ upon increasing the AuNPs concentration to 2 wt.% in SOPC MLVs. The reduction in the frequency of the C=O stretching mode generally indicates hydrogen bonding between carbonyl moieties of lipid and the surrounding particles, possibly through intramolecular hydrogen bonding and changes in the degree of hydration at the interfacial region of the lipid membrane. Severcan et al. [50] reported a similar trend, where the frequency of C=O stretching for pure DPPC MLVs at 1733 cm^−1^ shifted to 1730 cm^−1^ by increasing the melatonin concentration from 1 to 30 mol.%, indicating the possibility of hydrogen bonding either between the C=O groups of the lipids with the N–H group of melatonin or with the water molecules in the surrounding environment.

Another interesting band to probe changes near the lipid head group is the antisymmetric PO_2_^−^ group stretching vibration, which is sensitive to the hydration state of the lipid bilayers [51]. The antisymmetric PO_2_^−^ stretching vibration was observed at 1250 cm^−1^ for pure SOPC MLVs. As indicated in Table 2, no significant wavenumber shifts were noted in the antisymmetric PO_2_^−^ stretching of SOPC membranes in the presence of AuNPs up to 2 wt.%. On the whole, FTIR data revealed slight variations in methylene symmetric and antisymmetric stretch, but no major wavenumber shifts of C=O and PO_2_^−^ moieties, which indicates that the entrapped AuNPs did not show a major effect on altering the structure and conformation of acyl chains in the hydrophobic region and the lipid head groups at the interfacial region.

## 3. Materials and Methods

SOPC lipid was purchased from Avanti Polar Lipids Inc. (Alabaster, AL, USA). DPH, TMA-DPH, ethanol, heptane, and chloroform (purity 99%) were purchased from Sigma-Aldrich Chemie GmbH (Steinheim, Germany). All of the chemicals were used directly without any further purification. Dodecanethiol-stabilized AuNPs were purchased from nanoComposix Inc. (San Diego, CA, USA).

### 3.1. Liposome Preparation and Nanoparticle Characterization

SOPC MLVs were prepared by the thin film method as reported previously [52]. The SOPC lipid was dissolved in chloroform at a concentration of 5 mg/mL in a flask. The AuNPs were dispersed in heptane at a concentration of 1.25 mg/mL. These two mixtures were added in a calculated molar proportion to obtain the final concentrations of 0.5, 1, and 2 wt.% AuNPs in SOPC MLVs. The samples were evaporated under the vacuum for 4–5 h until the entire solvent was removed, and the dry lipid film was hydrated with double-distilled water and kept in a warm (~35 °C) ultrasonic bath above the gel-to-liquid phase transition temperature for at least 12 h to facilitate the formation of SOPC MLVs. Due to their extremely small size and hydrophobic coating, the AuNPs tended to be entrapped spontaneously at the hydrophobic region of the lipid bilayer membranes during the process of liposome formation. Pure SOPC MLVs (control) were also prepared following the same procedure without the addition of AuNPs. The morphology of AuNPs was observed using a JEOL 1010 transmission electron microscope (JEOL USA Inc., Peabody, MA, USA) operating at an accelerating voltage of 100 keV and an AMT XR41-B 4 MP (2048) bottom-mount CCD camera. The DLS measurements were performed using Zetasizer Nano ZS (Malvern Instruments, Malvern, UK) with a detection angle of 173°. A laser wavelength of 633 nm was used, and all the measurements were performed at a controlled temperature of 25 °C.

### 3.2. Fluorescence Spectroscopy

Temperature-dependent fluorescence anisotropy measurements were performed in a 10 mm pathlength cuvette using Cary Eclipse fluorescence spectrophotometer (Varian, Australia) at 25 °C, and using Varian Auto Polarizers with slits having a nominal band-pass of 5 nm for both excitation and emission. First, 2.5 μL of either DPH or TMA-DPH dissolved in 1 mM dimethyl sulfoxide (DMSO) solution was added to the SOPC MLVs to reach a final concentration of 1.0 μM DPH and 2.0 μM TMA-DPH in the total volume of 2.5 mL of SOPC MLVs. The anisotropy values were measured in the temperature range from 15 °C to 55 °C. The fluorescence anisotropy values were measured at an excitation wavelength of 358 nm with the excitation polarizer oriented in the vertical position, while vertical and horizontal components of polarized light were recorded through the monochromator at 410 nm for both probes. The anisotropy 〈r〉 values were calculated as shown in Equation (1) using the built-in software of the instrument.
(1)$$\langle r\rangle =\frac{I_{\left|\right|}-GI_{⊥}}{I_{\left|\right|}+2GI_{⊥}},$$
where $I_{\left|\right|}$ and $I_{⊥}$ are the emission intensities with polarizers parallel and perpendicular to the direction of the polarized exciting light, respectively. The values of the G-factor (ratio of the sensitivities of the detection system for vertically [I_HV_] and horizontally [I_HH_] polarized light) were determined separately for each sample. The lipid order parameter “*S*” was calculated from the anisotropy values by applying Equation (2) [53].
(2)$$S=\frac{{\left[1-2\left(\frac{r}{r_{0}}\right)+5{\left(\frac{r}{r_{0}}\right)}^{2}\right]}^{\frac{1}{2}}-1+\frac{r}{r_{0}}}{2\left(\frac{r}{r_{0}}\right)},$$
where *r*_0_ is the fluorescence anisotropy of DPH in the absence of any rotational motion of the probe, whose theoretical value is 0.4, while the experimental values generally lie between 0.362 and 0.394 [53]. In our calculation, the experimental value of *r*_0_ was 0.370.

### 3.3. FTIR Spectroscopy

For FTIR analysis, about 20 µL of each sample was spread on a transmission cell with silicon windows and air-dried before the analysis to avoid the interference of water signals. FTIR spectra were recorded using a Bruker VERTEX 70v with ATR attachment (Bruker Optik GmbH, Ettlingen, Germany). The FTIR spectra were measured in the 4000–600 cm^−1^ frequency region, accumulating 32 scans at a resolution of 4 cm^−1^. All measurements were repeated three times, and a similar trend was observed after each measurement.

## 4. Conclusions

Synthesis of liposome–AuNP hybrids represents a versatile nanocarrier system for drug delivery and biomedical applications. Therefore, it is important to study the influence of nanoparticles at various concentrations to understand their impact on the structure and diverse membrane properties such as fluidity, elasticity, and permeability. Considering this aspect, we analyzed the influence of different concentrations (0.5, 1, and 2 wt.%) of dodecanethiol-functionalized hydrophobic AuNPs on the structure and membrane fluidity of zwitterionic 1-stearoyl-2-oleoyl-*sn*-glycerol-3-phosphocholine (SOPC) lipid bilayer membranes using Fourier-transform infrared (FTIR) spectroscopy and fluorescent spectroscopy. Due to the small size (2.2 nm) and hydrophobic dodecanethiol coat, the AuNPs have a greater probability of being entrapped between the hydrophobic tail regions of lipids and interacting less with the hydrophilic head region of lipids. Accordingly, FTIR data indicated that the AuNPs did not cause a significant shift in the phosphate and carbonyl stretching vibrations of the SOPC lipids, which are sensitive to interactions of different particles near the polar head group region of lipids at the interfacial region. The fluorescent anisotropy measurements also showed that the incorporation of AuNPs at low concentrations of up to 2 wt.% in the vesicles did not show any notable changes in membrane fluidity. By combining the FTIR and fluidity data, we conclude that the hydrophobic AuNPs in the studied concentration did not adversely affect the structure and fluidity of the membrane, at least not within the studied percentage of the incorporated nanoparticles. On the other hand, if the size or amount of incorporated AuNPs is altered, it might cause a different effect.

Our findings indicate that the inclusion of AuNPs at a low concentration of up to 2 wt.% in liposomes can be considered to design safe AuNP–liposome hybrids for potential biological applications such as drug delivery and therapy. Since the morphology, charge, and surface chemistry of the nanoparticles play a major role in determining their interaction with the membrane and subsequent uptake by cells, more studies are certainly required to understand the different mechanisms via which nanoparticles cross the cell plasma membrane and induce cytotoxic effects. With these criteria in mind, further studies are underway to assess the safety of these particles on real cell membranes and their efficacy in drug delivery and cancer theranostics.

## Figures and Tables

**Figure 1 ijms-24-10226-f001:**
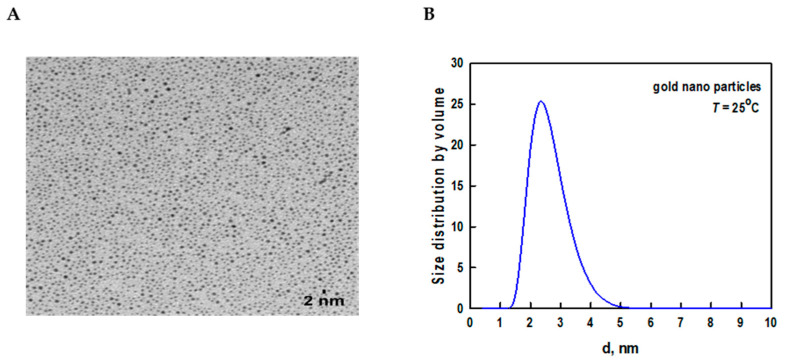
(**A**) TEM image of dodecanethiol-stabilized hydrophobic AuNPs. The scale bar corresponds to 2 nm. (**B**) Size distribution analysis of hydrophobic AuNPs by DLS.

**Figure 2 ijms-24-10226-f002:**
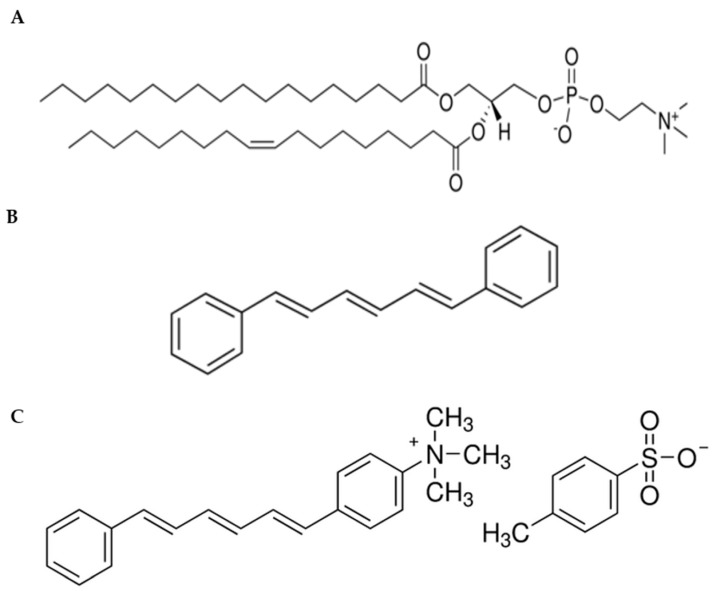
Chemical structures of (**A**) SOPC lipid, (**B**) 1, 6-diphenyl-1,3,5-hexatriene (DPH), and (**C**) 1-(4-trimethylammoniumphenyl)-6-phenyl-1,3,5-hexatriene (TMA-DPH) fluorescent probes.

**Figure 3 ijms-24-10226-f003:**
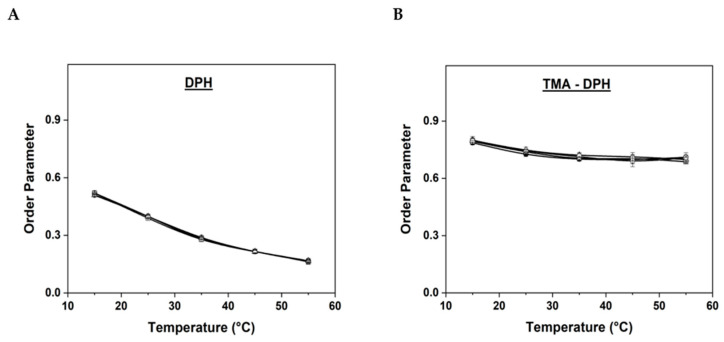
Lipid order parameter determined by (**A**) DPH and (**B**) TMA-DPH anisotropy measurements of SOPC MLVs with different concentrations of AuNPs (● pure SOPC MLVs taken as control; □ SOPC MLVs with 0.5 wt.% AuNPs; ◯ SOPC MLVs with 1 wt.% AuNPs; Δ SOPC MLVs with 2 wt.% AuNPs).

**Figure 4 ijms-24-10226-f004:**
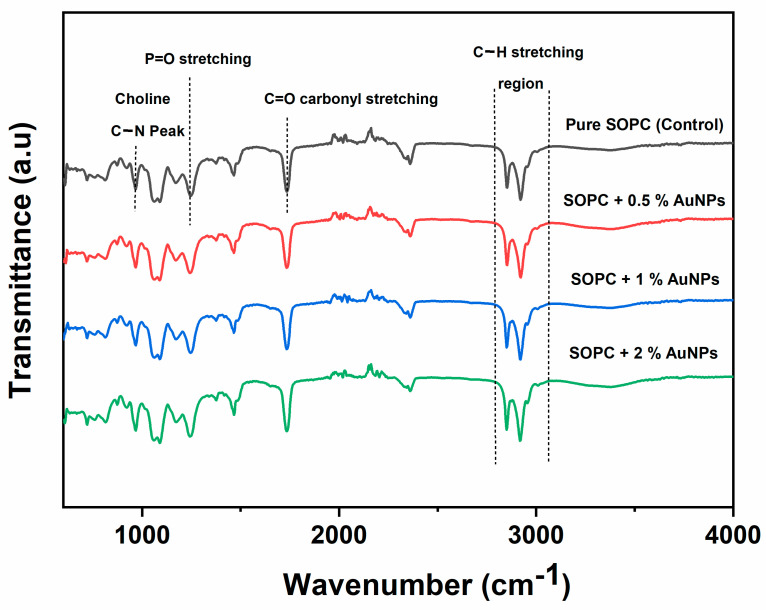
FTIR spectra of pure SOPC MLVs (control) and SOPC MLVs with 0.5, 1, and 2 wt.% AuNPs.

**Table 1 ijms-24-10226-t001:** Experimentally measured order parameter values of SOPC MLVs in the absence and presence of different concentrations of AuNPs (0.5, 1, and 2 wt.%), using DPH and TMA-DPH probes at different temperatures.

Temp (°C)	DPH				TMA-DPH			
	Pure MLVs	0.5% AuNPs	1% AuNPs	2% AuNPs	Pure MLVs	0.5% AuNPs	1% AuNPs	2% AuNPs
15	0.52 ± 0.01	0.519 ± 0.01	0.510 ± 0.01	0.51 ± 0.01	0.79 ± 0.01	0.79 ± 0.01	0.80 ± 0.01	0.80 ± 0.01
55	0.167 ± 0.01	0.162 ± 0.01	0.164 ± 0.01	0.167 ± 0.01	0.71 ± 0.01	0.68 ± 0.01	0.70 ± 0.02	0.69 ± 0.01

**Table 2 ijms-24-10226-t002:** Experimentally measured FTIR frequencies and wavenumber shifts (in cm^−1^) of SOPC MLVs in the absence and presence of different concentrations of AuNPs (0.5, 1, and 2 wt.%).

Samples	Choline C–H Stretching (cm^−1^)	Methylene C–H Antisymmetric Stretching (cm^−1^)	Methylene C–H Symmetric Stretching (cm^−1^)	C=O Stretching (cm^−1^)	PO_2_^−^ Antisymmetric Stretching (cm^−1^)	Choline C–N Peak Height Position (cm^−1^)
Pure SOPC	3011.0	2923	2850.0	1738.2	1250.0	969.5
SOPC + 0.5% AuNPs	3010.0	2922.5	2849.7	1738.0	1249.8	969.5
SOPC + 1% AuNPs	3009.6	2922.0	2849.1	1737.8	1249.5	969.3
SOPC + 2% AuNPs	3008.2	2921.0	2848.5	1737.3	1249.2	969.0

## Data Availability

No data is available due to privacy restrictions.

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
