# Peer review of "Effects of Hydrophobic Gold Nanoparticles on Structure and Fluidity of SOPC Lipid Membranes"

_ijms, 2023, doi:10.3390/ijms241210226_

Round 1

Reviewer 1 Report

In this work, the authors prepared and characterized functionalized gold nanoparticles with dodecanethiol and 1-stearoyl-2-oleoyl-sn-glycerol-3-phosphocholine lipid. Results demonstrate that the nanoparticles do not cause significant alterations in the structure and membrane fluidity. This is interesting for biomedical applications. The manuscript is clear and detailed, the conclusions are in accordance with the achieved results, therefore, it can be considered for publication if the following revision is considered:

-        In line 133, where is 0.5197 ± 0.01, should be replaced 0.52 ± 0.01,

-        In line 134, where is 0.5194 ± 0.003, 0.51 ± 0.003, and 0.5087 ± 0.01, should be 0.519 ± 0.003, 0.510 ± 0.003, and 0.51 ± 0.01.

-         Similar adjustments should be made to the values of the 137, 138 and 194 lines.

-        If possible, include a table with the order parameter values.

The nanuscript is clear but the English should be revised.

Reviewer 2 Report

Ref_comments to the paper titled as “Effects of hydrophobic gold nanoparticles on structure and fluidity of SOPC lipid membranes” written by the authors:

Poornima Budime Santhosh, Tihomir Tenev, Luka Šturm, Nataša Poklar Ulrih and Julia Genova

Despite of a large number of works in the synthesis, characterization, applications of the gold nanoparticles, the investigation of their features in the biomedicine area is actual and modern. From this point of view the current article is actual and modern.

For the first. The authors have made nice literature search connected with the analysis of 53 references. The manuscripts published last 5 years have been included and analyzed in this consideration as well. It is confirmed that the authors are known the area of study with good advantage.

For the second. The paper has good illustrations. The size distribution analysis of the gold nanoparticles has been made; chemical structure of the lipid and lipid order parameter has been presented; Furrier spectrometer data have been established; some analytical calculations have been made.

Some questions are the followings:

1). You have determined the lipid order parameters. Good. Have you some data about the refractive index change from the pure lipid to the lipid structures with the gold NPs?

2). You have calculated the anisotropy values via Eq. 1. Have you determine the graphic data about the transparency of your doped composite materials for the parallel and orthogonal light components taken into account the comparison between pure materials and doped one?

3). What is about the stability of your lipid with gold NPs?

Conclusion are extended.

As for my local opinion, this paper can be published after minor corrections regarded to the questions..

Round 2

Reviewer 1 Report

This manuscript has been improved considering the suggestions and, therefore, I can recommend this work for publication.